# IoT-Based Resource Control for In-Vehicle Infotainment Services: Design and Experimentation

**DOI:** 10.3390/s19030620

**Published:** 2019-02-01

**Authors:** Dong-Kyu Choi, Joong-Hwa Jung, Ji-In Kim, Moneeb Gohar, Seok-Joo Koh

**Affiliations:** 1School of Computer Science and Engineering, Kyungpook National University, Daegu 41566, Korea; supergint@gmail.com (D.-K.C.); godopu16@gmail.com (J.-H.J.); 2Silla System Corporation, Daegu 41065, Korea; jiin16@gmail.com; 3Department of Computer Science, Bahria University, Islamabad 44000, Pakistan; moneebgohar@gmail.com

**Keywords:** In-Vehicle Infotainment (IVI), Internet of Things (IoT), Resource Control, Master

## Abstract

A variety of in-vehicle infotainment (IVI) devices and services have been developed by many vehicle vendors and software companies, which include navigation systems, cameras, speakers, headrest displays, and heating seat. However, there has not been enough research on how to effectively control and manage numerous IVI resources (devices and contents), so as to provide users with more enhanced services. This paper proposes a framework of resource control for IVI services so as to efficiently manage the IVI resources within an automobile. Differently from conventional IVI systems, in the proposed scheme, the *IVI-Master* is newly introduced for overall control of IVI resources, and IVI users are divided into *owner* and *users*. In addition, the IVI resources are classified as *personal resources* and *shared resources*, which are managed by the IVI-Master using the Lightweight Machine-to-Machine (LWM2M) standard. The proposed IoT-based IVI resource control scheme was implemented and tested. The experimental results showed that the proposed scheme can be used to effectively manage IVI resources for users. Additionally, the proposed resource control scheme shows lower bandwidth usage than the existing scheme.

## 1. Introduction

As automobile and IT industries continue to grow, autonomous vehicles have now evolved from a simple transportation means into cultural and living spaces by changing individuals’ lifestyle. With this trend, many in-vehicle infotainment (IVI) services [1,2,3] and devices have been developed by numerous automobile manufacturers, as illustrated by the examples of Mercedes-Benz, Toyota, BMW, and General Motors [4,5,6,7]. *Infotainment* is a new compound word combining *information* and *entertainment*. Such infotainment devices include navigation systems, cameras, speakers, headrest displays, air-conditioners, thermometers and heating seats, and lights.

It is also noted that a variety of IVI services are being developed by global software companies, such as Google and Apple [8,9], and the associated issues have been discussed by the GENEVI consortium [10]. In addition, many IVI services and products are demonstrated in the Consumer Electrics Show (CES) event. For example, the BMW i3 is equipped with automatic parking and collision avoidance technologies, and four laser scanners are mounted on the car to monitor the surrounding environment and prevent traffic accidents [11]. It is expected that a variety of IVI devices and services will be developed in the future. Newer vehicles will provide a wide range of systems that allow devices (e. g., smartphones and laptops) to be connected to a variety of services embedded within the vehicle.

In the meantime, the Open Connectivity Foundation (OCF) [12] and Open Mobile Alliance (OMA) [13] are defining the standards associated with automobiles. They mainly focus on ways to efficiently connect automobiles to the existing communication infrastructures, known as vehicle-to-vehicle (V2V), vehicle-to-infrastructure (V2I), and vehicle-to-everything (V2X) communication. The Internet Engineering Task Force (IETF) is also discussing the automotive and Internet connectivity in the IP Wireless Access in Vehicular Environment (IPWAVE) Working Group [14], which focuses on how to provide external communications for autonomous smart cars, such as V2V, V2I, and V2X communication.

Even though many commercial products for IVI services have been developed by automobile manufacturers and software companies, few studies have examined how to effectively provide such IVI services. The existing works have mainly focused on the adaptive user interface to increase the user’s convenience in the IVI system, such as voice recognition or electronic secretary [15,16,17,18]. Some works have examined the concept of IVI system or challenges/requirements to implement it [19,20]. 

Therefore, there is a crucial need to study how to effectively control and manage a variety of IVI resources (devices and contents) within an automobile, so as to provide IVI users with more enhanced infotainment services. In the conventional IVI system, only the vehicle owner is considered, while other users are not. That is, only the owner can use and control each of the IVI devices directly; thus, there is no unified framework to control the IVI resources.

In this paper, we propose a framework of resource control for IVI services, based on Internet-of-Things (IoT) [21,22,23,24] technology, to efficiently manage the IVI resources within an automobile. The proposed framework has the following features: 1) introduction of the IVI-Master for overall control of IVI resources and interaction with users, 2) consideration of other types of users as well as the vehicle owner for IVI services, 3) classification of personal and shared resources, and 4) use of the Lightweight Machine-to-Machine (LWM2M) standard [25,26,27] for IoT communications between IVI-Master and resources. The IVI-Master is responsible for management of all IVI resources and will interact with the owner and/or the users. With the help of LWM2M, the IVI-Master can efficiently manage a large number of IVI resources. The owner can add and remove IVI resources (all devices and contents in the vehicle) to the IVI-Master. The users can control and use the IVI resources as per their pre-specified authority or permission level. 

The proposed scheme is based on the IVI-Master for centralized control of IVI resources, differently from the existing peer-to-peer resource management model. With the help of the IVI-Master and IoT technology, we can provide differentiated IVI services for each user and reduce the traffic load by maintaining the user context. To the best of our knowledge, this paper is the first proposal of IoT-based centralized control of IVI resources. 

This paper is organized as follows. In Section 2, we describe the framework of the proposed IoT-based resource control for IVI services. Section 3 presents the operations for user registration and resource control. In Section 4, we discuss the prototype implementation and experimental results for the proposed scheme.

## 2. Framework of IoT-based Resource Control

### 2.1. System Architecture

Figure 1 shows the architecture of the existing IVI system, in which a user (vehicle owner) controls and uses numerous IVI resources through one-to-one communication. The user will exchange the request and response directly with the devices or contents. Thus, it is not easy for a user to control and manage many different types of IVI resources, since the distinctive features of each resource must be considered by the user. In addition, only the owner is considered for IVI services, and the types of users (e. g., family, friends, and public users) are not supported. Depending on the user type, different services and permission levels for IVI resources may be provided. Furthermore, each IVI device may use a different communication technology, such as Bluetooth, ZigBee, WLAN, etc.

In the meantime, Figure 2 shows the proposed IoT-based IVI system architecture. The IVI-Master is newly introduced for overall control and management of various IVI resources, such as sensors, devices, and contents. The IVI users are classified as vehicle owner or other users. The vehicle owner will manage the IVI-Master and all IVI resources with the associated database (DB). The other users will use and control the IVI resources with their authority and permission level with the help of the IVI-Master. The communications between the IVI-Master and users will be done by using the HTTP, whereas LWM2M is used for communications between the IVI-Master and IVI resources, in which CoAP (Constrained Application Protocol) [28] and/or Message Queuing Telemetry Transport (MQTT) [29] may be used.

Figure 3 shows a possible configuration of IVI system components, based on Figure 2, which includes the IVI-Master, user, and many IVI resources.

Table 1 shows the differences between the existing and proposed IVI systems. From the viewpoint of network topology, the existing system is based on a one-to-one (peer-to-peer) connection between the owner and each resource, whereas the proposed system employs a star topology between the IVI-Master and many resources. In the existing scheme, only the vehicle owner is considered, whereas users are classified into the owner and the other users in the proposed scheme. In the existing scheme, each resource is controlled by the owner directly, while in the proposed scheme the IVI-Master is employed for overall control of many IVI resources. Resources are classed as personal and shared in the proposed scheme, with personal resources being used by a particular user, and shared resources by one or more users. In the existing scheme, the MAC/PHY protocols are used for communication, whereas IoT-based LWM2M protocols, such as CoAP, MQTT, and HTTP, are used in the proposed scheme.

### 2.2. Types of Resources: Personal and Shared 

IVI resources represent sensors, devices, and contents for IVI services. In this paper, IVI resources are classified into *personal* and *shared resources*. Personal resources are referred to as resources that can be used by a particular user, which may include individual seats, a headrest display, a heated seat, and a cooled seat. on the other hand, shared resources can be used by one or more users, which may include the device to monitor the inner states of the vehicle, such as temperature, humidity, oil level, battery, etc. Each resource will be classified as a personal or shared resource in the system initialization, and the list of resources available for each user will be managed by the IVI-Master.

In addition to devices, multimedia contents are also regarded as IVI resources, which may include video, traffic information, music, etc. Such contents can be displayed for users under the control of the IVI-Master. The MQTT protocol may be used for delivery of multimedia contents.

### 2.3. Types of Users: Owner and Other Users

In the existing scheme, only the owner is considered as a user of IVI services. However, a variety of users may be considered. For example, IVI services may be used for business purposes, such as car-sharing services, rental services, public transportation, etc., and family members and friends may also use the IVI services. Accordingly, in the proposed scheme, users are divided into the *vehicle owner* and *other users*. Further classification may also be possible, which is a topic for further study.

The owner performs the overall control and management of all IVI resources. In the meantime, users have their specific authority and permission level for each resource, which may be negotiated and determined in the user registration process.

### 2.4. Use of LWM2M for Communication

There are numerous kinds of devices in an automobile, and each may use its distinctive hardware and communication protocol, such as Bluetooth, ZigBee, and WLAN. In addition, new devices and sensors may be added to the automobile. Therefore, a unified communication method is required to support many different types of IVI devices. In the proposed scheme, a IoT-based LWM2M framework is used for communication between the IVI-Master and IVI resources, which is a well-known IoT device management framework suitable for mobile platforms.

The LWM2M standard was developed by the Open Mobile Alliance (OMA), and uses the application layer communication protocol (e.g., CoAP) between the LWM2M server and the LWM2M client for IoT device management, as shown in Figure 4. The LWM2M client represents a device to be managed, and the LWM2M gateway or server will be used to manage the devices. In the figure, the LWM2M client sends a POST request to register itself with the LWM2M server. The server will verify the request, and then registers the client, creates a resource for managing the client, and returns the resource URL to the client. If the client does not receive the control message for a certain period of time, it will operate in off-line mode. At that time, if the server needs to send a command to the client, the server will store the command in its own queue. After a certain sleep time, the client will send a POST message to the resource URL of the registered server to inform that it is operating in on-line mode. The server that receives the message sends the command stored in its queue.

In this paper, the LWM2M is slightly extended to support the IVI-Master and IVI resources, as shown in Figure 5. The figure shows an example of device registration and management procedures in the proposed scheme. The LWM2M procedures in Figure 5 are almost the same as those in Figure 4, but are extended to support the distinctive features of IVI services, such as the IVI-Master, IVI resources, and users. In the figure, the IVI-Master receives a registration request message from the IVI resource, and waits for permission from the owner. The IVI-Master then sends a response to the IVI resource. More detailed operations for IVI services will be described in the next section.

## 3. Operations for User Registration and Resource Control

### 3.1. User Registration

Figure 6 shows the registration operations for the owner. If the IVI-Master does not have the information of the owner, it broadcasts its URL information to the network. The user who wants to register as the owner should send a POST message to the IVI-Master. When the IVI-Master receives the POST message, it will register the owner in its DB and then respond with the associated authentication key. The user (owner) must then send a PUT message with the authentication key to the IVI-Master. The IVI-Master will check the authentication key and send the 200 OK response message to complete the registration process.

Figure 7 shows the registration and log-in procedures for another user, in which it is assumed that the user is still not registered with the IVI-Master. The user will first send a POST message that contains the user information. Upon receiving this message, the IVI-Master generates a URL for authorization and waits for the owner’s permission. At this time, the IVI-Master can notify the owner by using MQTT. The owner can grant permission to the user by sending a PUT message containing the authority level to the generated URL. The IVI-Master returns the authentication key to the user with the 201 Created HTTP response. The user who receives the authentication key can now log in by sending a PUT message with the received authentication key.

### 3.2. Resource Control for Owner

Figure 8 shows the procedures for requesting the IVI resource list from the owner. The log-in process is omitted in this figure, as it was already described in the user registration process. The owner does not require any additional authentication procedure to perform these operations. The owner can just receive the IVI resource list by simply sending a GET request to the IVI-Master.

As shown in Figure 9, the owner can also control the IVI resources by sending a GET or PUT request message with the associated URL to the IVI-Master. For energy efficiency, our scheme is designed based on the queuing model of LWM2M. That is, if the IVI resource is in the off-line mode, the IVI-Master will store the instructions in its buffer until it receives the POST message from the IVI resource. After receiving the POST message, the IVI-Master will transmit the next command.

### 3.3. Resource Control for Users

Figure 10 shows the operations of IVI resource list request for other users. The user must obtain permission from the owner each time he/she wants to receive the IVI resource list. The user requests the resource list from the IVI-Master by using the GET method. Subsequently, the IVI-Master creates a URL for the user authentication. The owner sends a message to the IVI-Master containing the authority level for each resource. 

Figure 11 shows the operations for control of personal resources by user. The user transfers the command to the IVI-Master by using the PUT method. The IVI-Master waits until the resource is in on-line status. If the on-line status is confirmed, the IVI-Master will forward the command stored in the queue to the resource. It is noted that the user does not require the owner’s permission to control the personal resource. 

Figure 12 shows the resource control for shared resources, in which the user sends the command to the IVI-Master by using the PUT method. The IVI-Master then identifies the type of the corresponding IVI resource and generates a URL for authentication in case of a shared resource. After the IVI-Master receives the owner’s permission message, it transfers the command stored in the queue to the IVI resource when the shared resource is available to use. That is, the control of shared resources needs the owner’s permission.

## 4. Performance Analysis and Experimentations

### 4.1. Analysis by Simulation

First, we analyzed the performance of the proposed scheme by simulation from the viewpoint of the data volume generated to control the IVI resources. For comparison, we derived the numerical equations by analysis, and used the *Apache Commons Math Library* [30] to measure the volume of data in the simulation. Table 2 shows the parameters used in simulations, in which T represents the simulation time.

In the table, I_comm_ represents the time interval for generation of messages to be sent to a sensor, which is generated as per the exponential distribution with an average of T/3 second. M_con_, M_creq_, and M_cres_ indicate the sizes of the messages for connection, command request, and command response, respectively, which are configured based on the conventional IoT messages. P is the probability that two or more commands will be delivered to a sensor when the sensor is in sleeping mode. In this simulation, P was set to 0.15. T_cycle_ is the period of the duty cycle of one IVI resource, which is set to 1/2 of the simulation time T. N_sensor_ and N_comm_ represent the numbers of IVI resources and commands transferred to IVI resources, respectively.

Subsequently, we compared the volume of data generated for the existing and proposed schemes, as the number of IVI resources increases. The volumes of data in the existing scheme ($V_{e})$ are calculated based on the equation (1), and those in the proposed scheme $\left(V_{p}\right)$ are calculated based on the equation (2). In the equation, $N_{comm}$ is randomly generated by Poisson distribution where the average (λ) is 3$\cdot N_{sensor}$.
(1)$$V_{e}=N_{comm}\cdot \left(M_{con}+M_{creq}+M_{cres}\right)$$
(2)$$V_{p}=N_{sensor}\cdot \frac{T}{T_{cycle}}+N_{comm}\cdot \left(1-P\right)\cdot \left(M_{creq}+M_{cres}\right)$$
Figure 13 shows the simulation results, in which T was set to 60 seconds. In this figure, it is shown that the data volume generated in the proposed scheme is smaller than the data volume in the existing scheme by approximately 15%. This is because the proposed scheme can reduce the generated traffic by using the IVI-Master and queuing model. 

### 4.2. Implementation and Testbed Experimentations

In this section, we discuss the prototype implementation of the proposed scheme. Figure 14 shows the associated testbed configuration, in which we used the *Latte-panda* board [31] as the IVI-Master, and *Raspberry pi* and *Arduino shield* as IVI resources. Table 3 summarizes the specifications of the equipment and software used for implementation and testbed experimentations. Communications between the IVI-Master and IVI resources are made based on the Wi-Fi technology.

Figure 15a shows the screen captured when the owner sends a request to receive the IVI resource list, and Figure 15b shows the screen captured when the user sends a request to receive the resource list. The owner does not need to go through an authentication process because he/she already has the authority to access the resource list, but the user can receive the resource list only with the permission of the owner.

Figure 16 shows the packet capturing results for the resource list request by the user. In (1), the user sends a POST message to request the registration, and the PUT message of the owner shows the process of authorizing the request. After that, the user sends a PUT message with the authentication key received in response to the IVI-Master in (2), and requests the resource list, as shown in (3). However, since the user does not have access authority to the resource list, the IVI-Master creates a resource for permission requests (*/res/permission/AweUkf16Gw*). The owner sends a PUT message to the URL of the resource created to allow it in (4), and the IVI-Master sends the response message to the user along with the IVI-Resource list, as shown in (5).

Figure 17a shows the screen capture associated with the shared resource request by the owner, and Figure 17b shows the same for the user. The owner can use the shared resource without authentication. The user, however, must be authorized by the owner. In this experimentation, we used a speaker as the shared resource.

Figure 18 shows the packet capturing results when the user requests a shared resource. First, the user sends a GET message to use the resource in (1). In this experiment, GET and PUT messages were used to request the resources of speaker and display, and a PUT message was used for the resources of air conditioner or heated seat.

Figure 19 shows the case of the personal resource request by owner and user. Unlike shared resources, in this case, both owner and user can use their personal resources without authentication. In this experiment, the headrest displays for each seat were used as personal resources. 

### 4.3. Performance Comparison by Exprimentation

The proposed scheme uses the IVI-Master for resource management to alleviate the traffic load between IVI-Master and IVI resources, whereas the existing scheme is based on the direct control of IVI resources by the owner. For a performance analysis, we compared the volume of data for the existing direct control scheme and the proposed scheme.

Figure 20 shows the experimental results. In this experimentation, we measured the volume of data generated by 50 processes, with each process performing the functionality of *Raspberry pi*. As seen in the figure, the data volume of the proposed scheme was smaller than that of the existing scheme. The gap in performance increases as the number of resources increase. It is noted that these results are almost the same as the simulation results shown in Figure 13. The only difference is that the variations in data volume in the experimentation are slightly larger than those in the simulation. 

We compared the traffic density of each candidate scheme by experimentation time (Figure 21 and Table 4). The traffic density is measured by the number of packets generated in 10 minutes. In Figure 21, the proposed scheme generates traffic with a relatively uniform distribution over experimentation time, compared with the existing scheme. Table 4 shows the average and standard deviation values of the bandwidth used in the existing and proposed schemes. The proposed scheme was found to use the network bandwidth more effectively and stably than the existing scheme. This is because the proposed scheme can control the network traffic in a timely manner by using the IVI-Master for control of IVI resources. That is, the proposed scheme can prevent the overloading of control messages.

In the meantime, Figure 22 shows the response time for each request message as the number of IVI resources increases. In this figure, we can see that the response times for the proposed scheme are shorter than those for the existing scheme. This is because all resources are controlled by using the IVI-Master in the proposed scheme, whereas a user directly controls the resources in a one-to-one manner in the existing scheme.

## 5. Conclusions and Future Research

Many types of IVI services and devices have recently been developed, and it is expected more of such devices and services will be newly developed in the future. Accordingly, there is a crucial need to study how to effectively control and manage a variety of IVI resources, so as to provide IVI users with more enhanced infotainment services. 

In this paper, we proposed a unified framework of resource control for IVI services, so as to efficiently manage the IVI resources within an automobile. In the proposed scheme, the IVI-Master is employed for overall control of IVI resources and interaction with users, and general users are considered as well as the vehicle owner. In addition, the IVI resources are classified into personal and shared resources, and the LWM2M standard is used for IoT communications between the IVI-Master and resources. The experimental results showed that the proposed scheme can be used to effectively manage the IVI resources for owner and users. Additionally, the proposed resource control scheme shows lower bandwidth usage than the existing scheme.

This paper focused on the design of a framework for IoT-based IVI resource control, and the validation of the proposed scheme by experimentation. In future work, we will investigate a more detailed design of IVI control operations for different resource and service types, in which we may need to consider non-critical services (such as navigation system, speakers, etc.) as well as critical services (such as vehicle sensors, rear view camera, and so on). In addition, we will consider security issues associated with IVI services, which include illegal access of external users and authentication/authorization procedures for IVI resources.

## Figures and Tables

**Figure 1 sensors-19-00620-f001:**
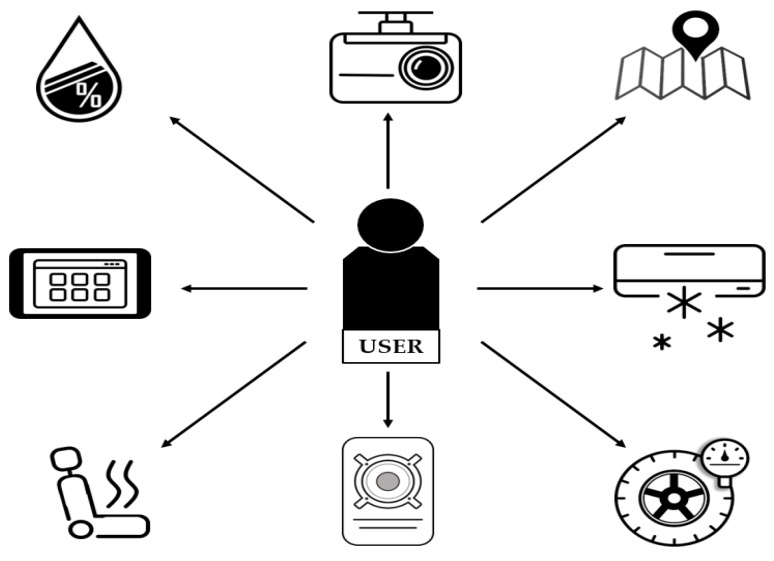
Architecture of the existing in-vehicle infotainment (IVI) system.

**Figure 2 sensors-19-00620-f002:**
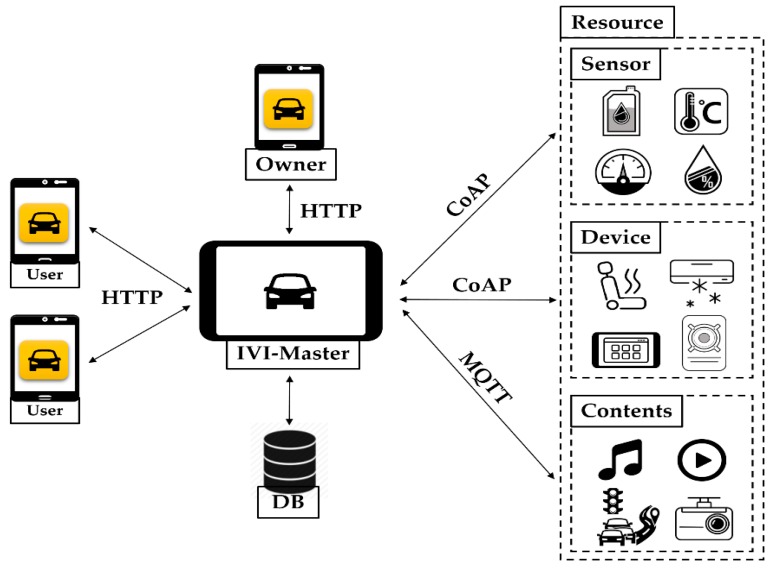
Architecture of proposed IVI system.

**Figure 3 sensors-19-00620-f003:**
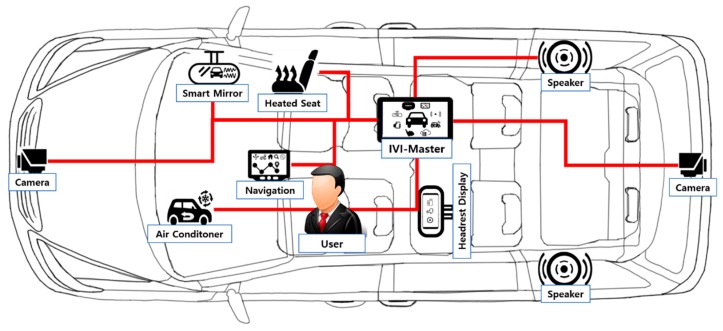
Configuration of proposed IVI system components.

**Figure 4 sensors-19-00620-f004:**
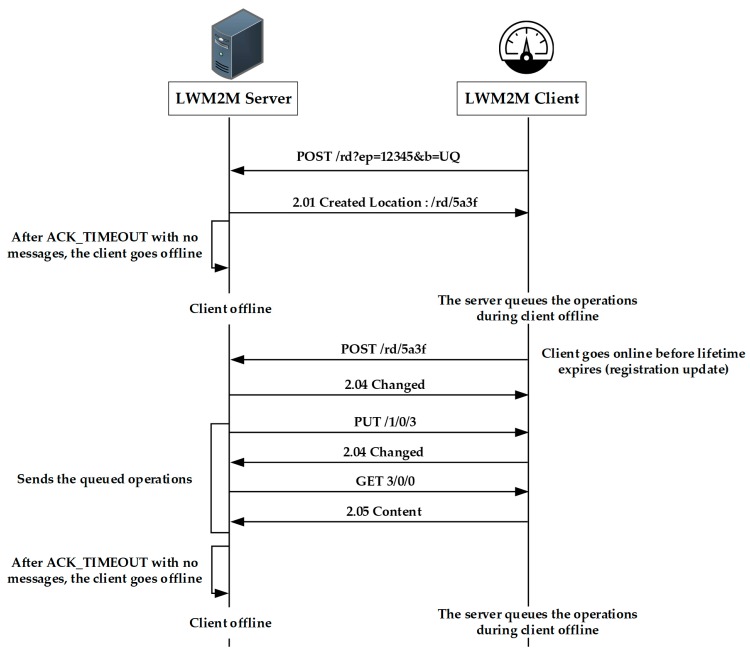
Lightweight Machine-to-Machine (LWM2M) operations.

**Figure 5 sensors-19-00620-f005:**
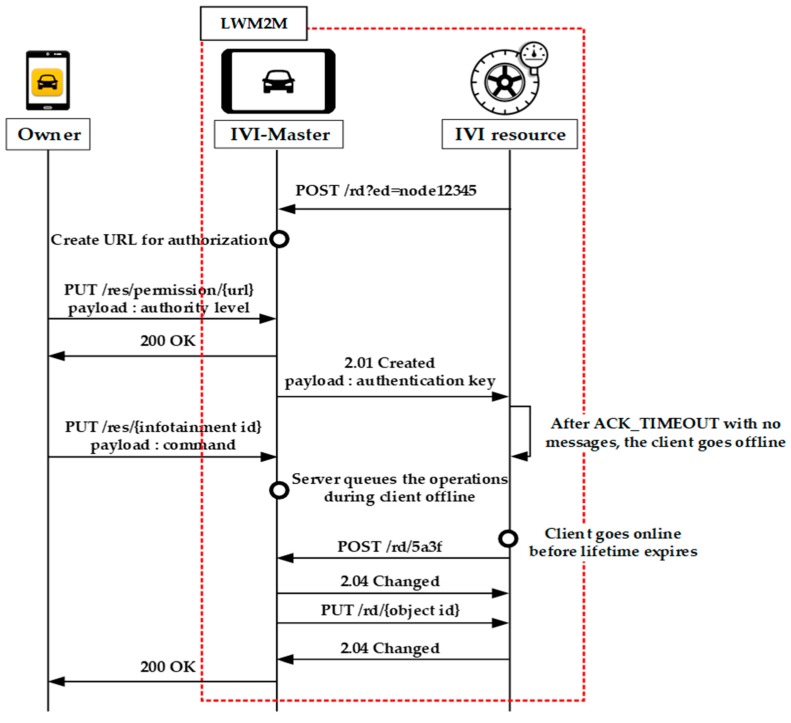
Use of LWM2M standard for IVI services.

**Figure 6 sensors-19-00620-f006:**
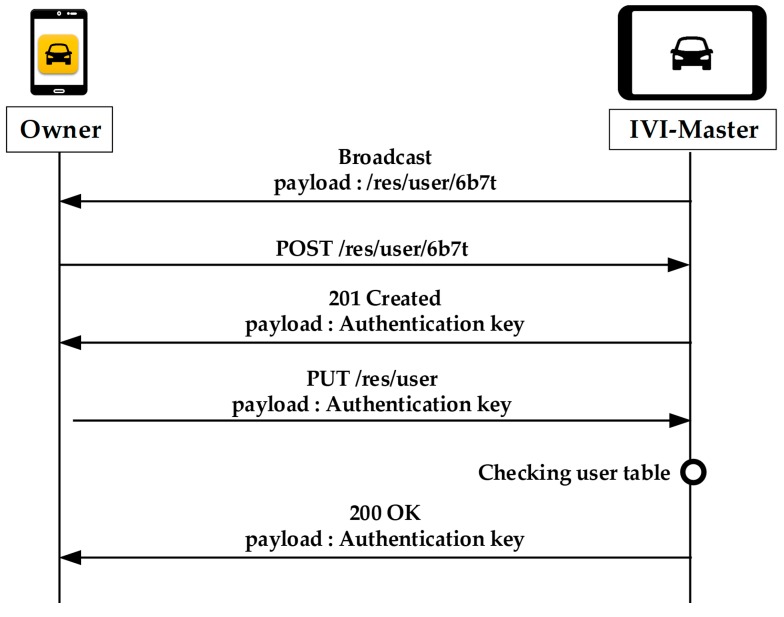
Registration of the owner.

**Figure 7 sensors-19-00620-f007:**
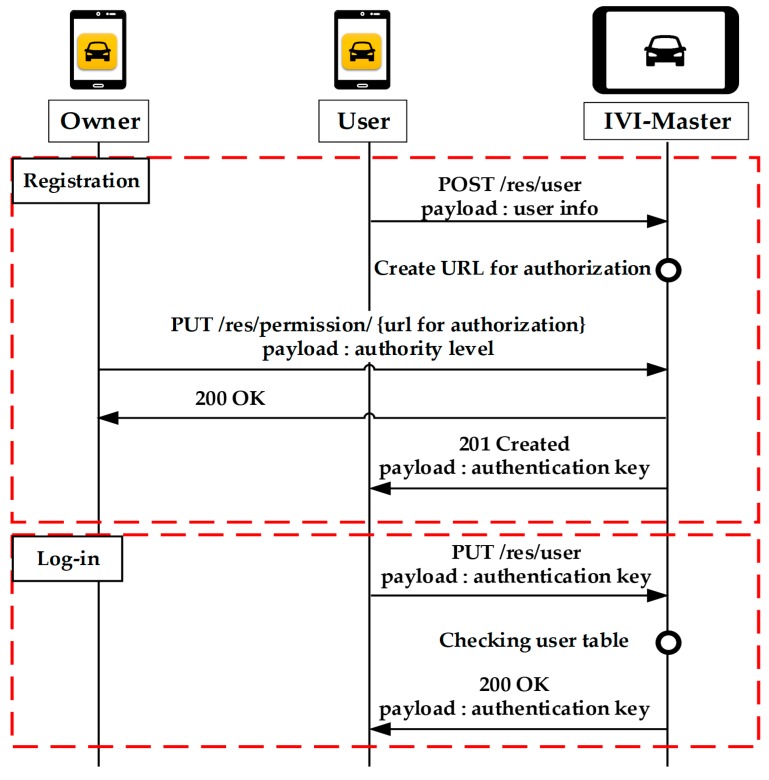
Registration of user.

**Figure 8 sensors-19-00620-f008:**
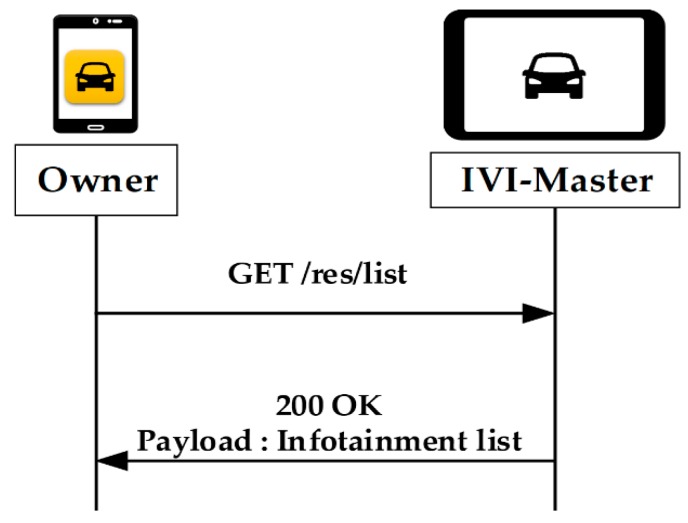
Request of the resource list by owner.

**Figure 9 sensors-19-00620-f009:**
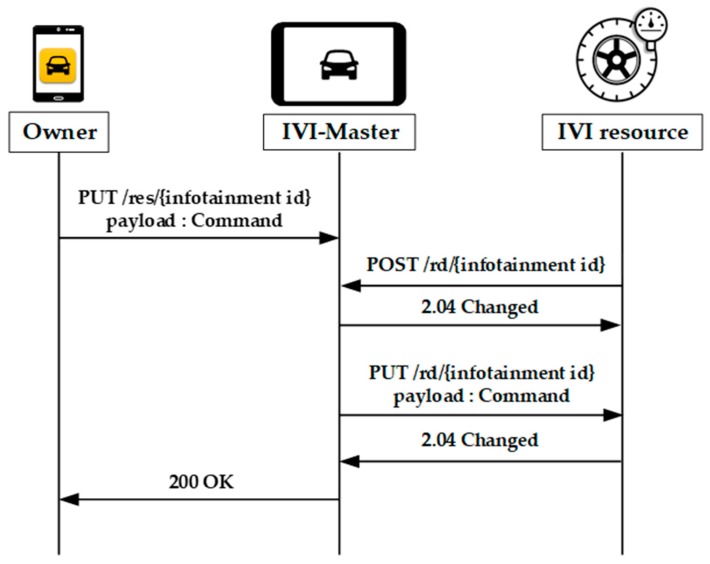
Resource control for owner.

**Figure 10 sensors-19-00620-f010:**
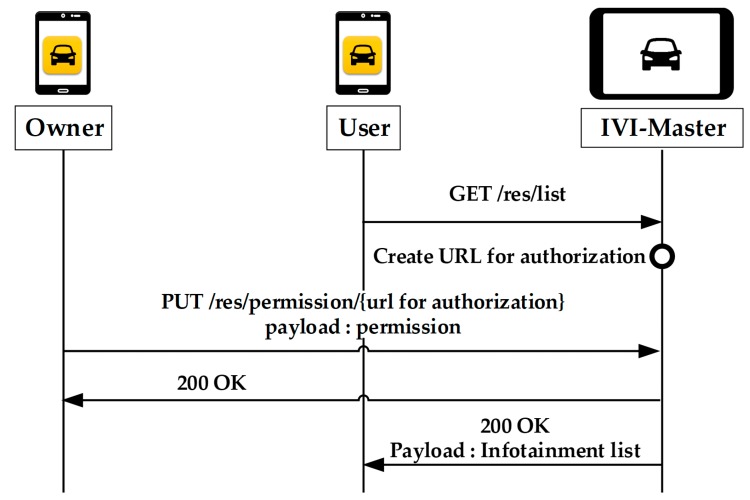
Request of resource list by user.

**Figure 11 sensors-19-00620-f011:**
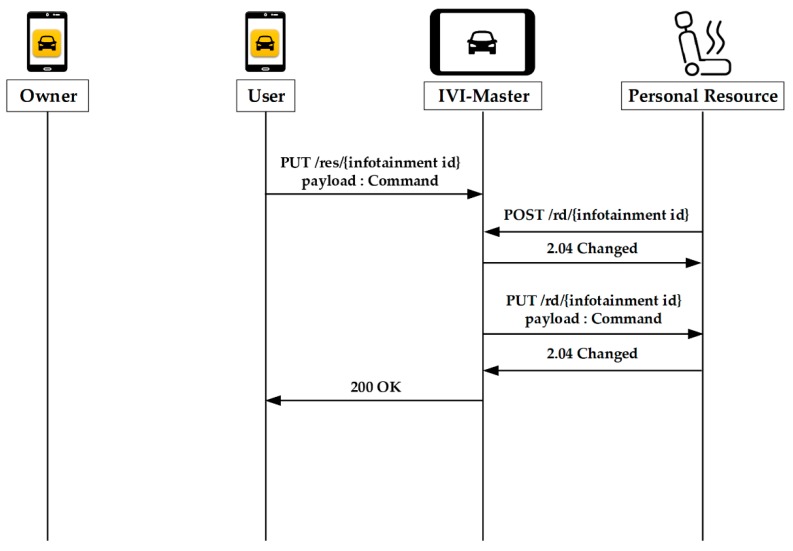
Control of personal resources by user.

**Figure 12 sensors-19-00620-f012:**
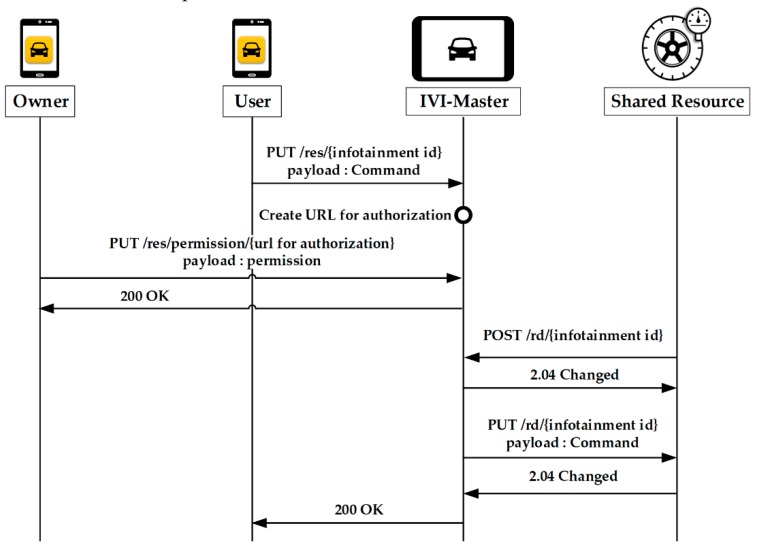
Control of shared resources by user.

**Figure 13 sensors-19-00620-f013:**
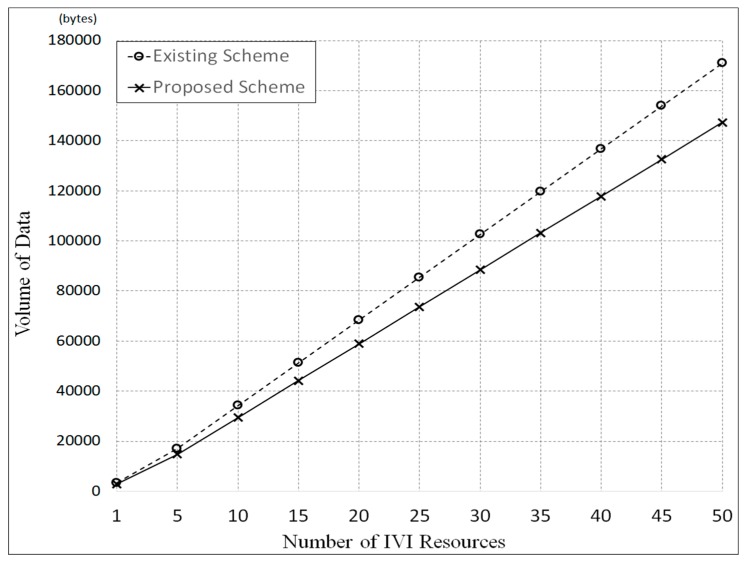
Comparison of data volume used in IVI resources control by simulation.

**Figure 14 sensors-19-00620-f014:**
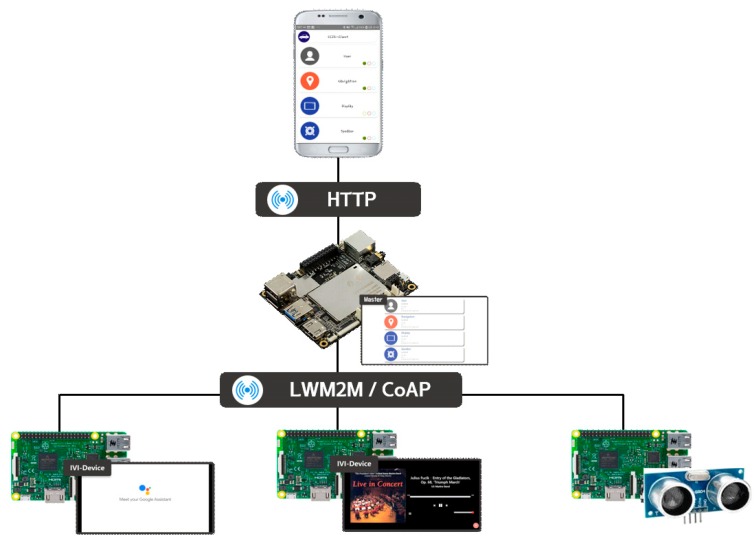
Testbed configuration.

**Figure 15 sensors-19-00620-f015:**
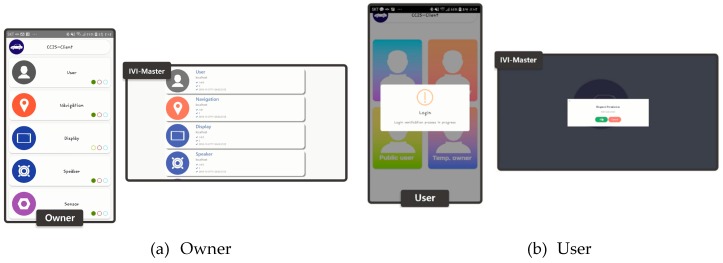
Screen captures for resource list requests from owner and user.

**Figure 16 sensors-19-00620-f016:**
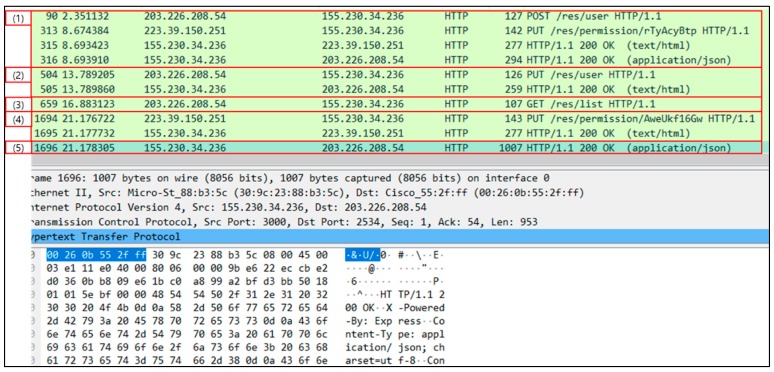
Packet capturing results for resource list request by user.

**Figure 17 sensors-19-00620-f017:**
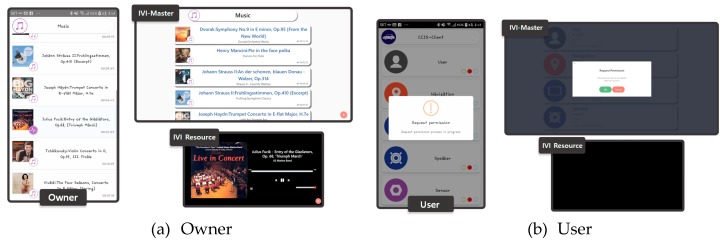
Screen captures for shared resource control by owner and user.

**Figure 18 sensors-19-00620-f018:**
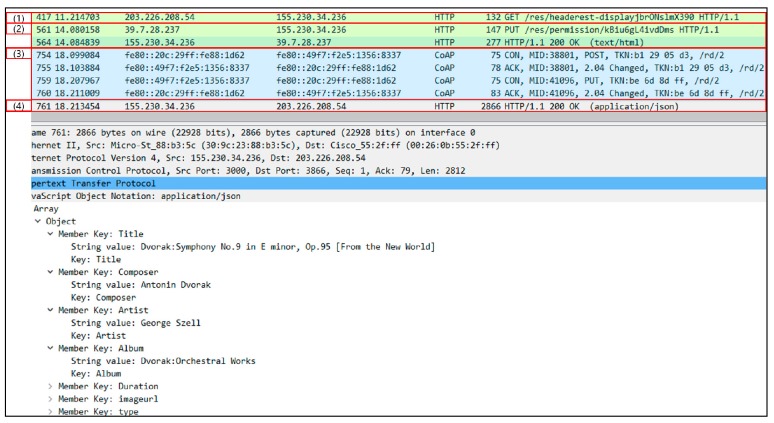
Packet capturing results for shared resource control by user.

**Figure 19 sensors-19-00620-f019:**
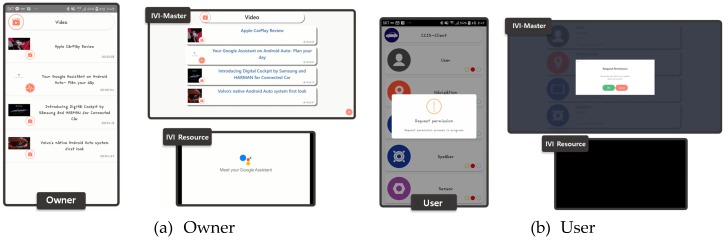
Personal resource control by owner and user.

**Figure 20 sensors-19-00620-f020:**
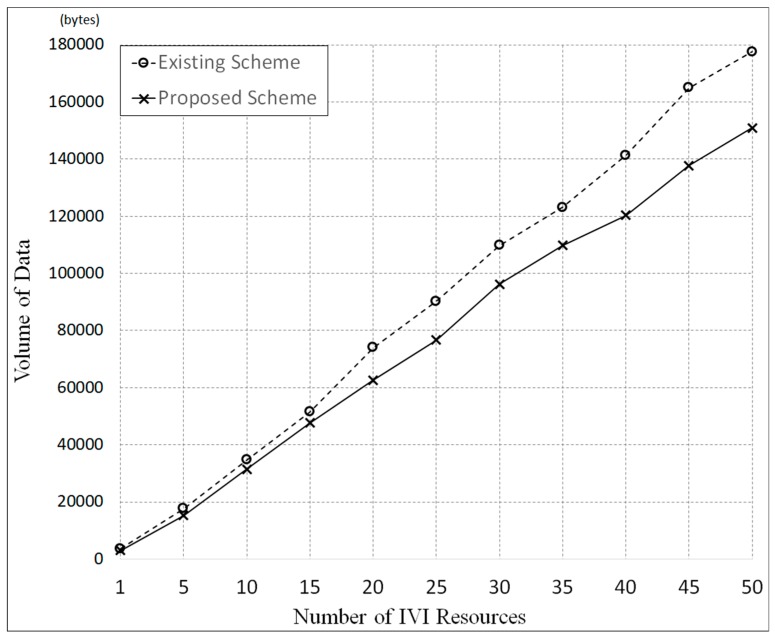
Comparison of data volume used for IVI resources control by experimentation.

**Figure 21 sensors-19-00620-f021:**
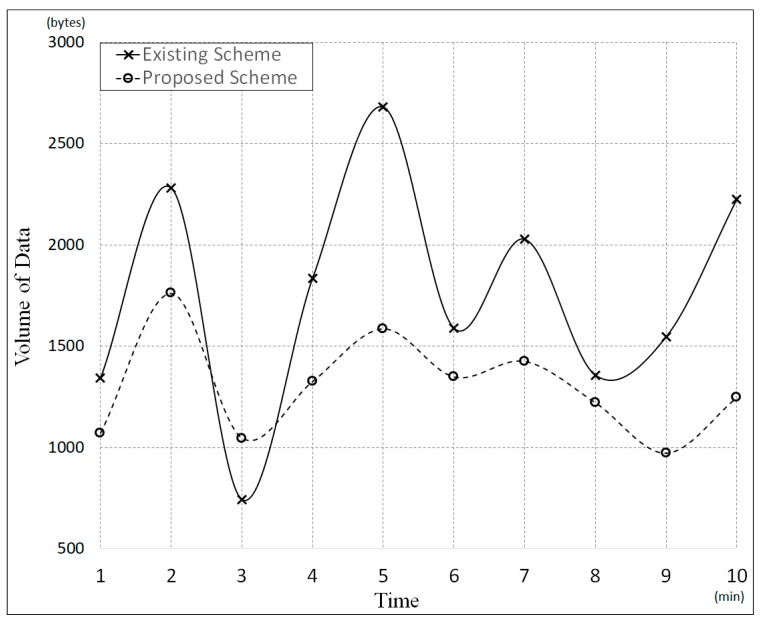
Comparison of bandwidth usage during experimentation.

**Figure 22 sensors-19-00620-f022:**
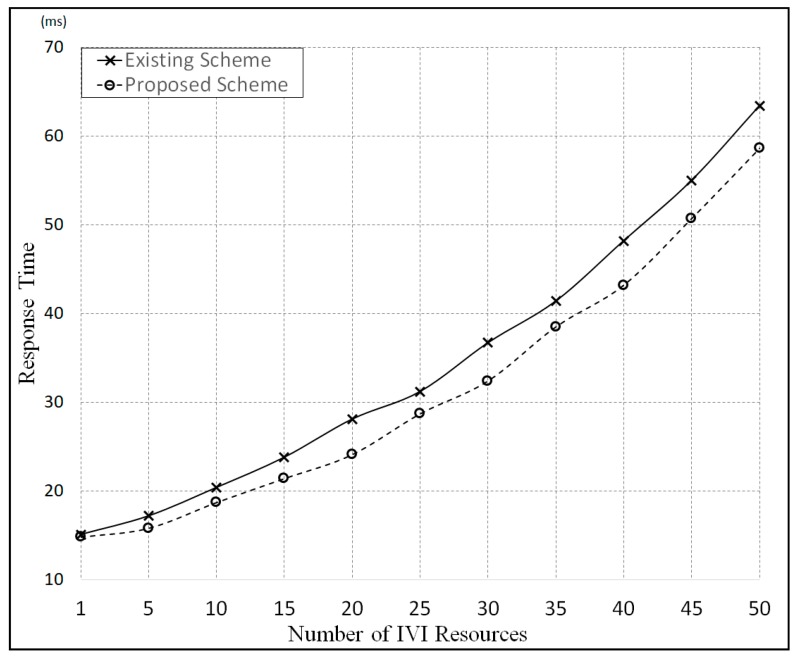
Comparison of response times for IVI resource control messages.

**Table 1 sensors-19-00620-t001:** Architectural comparison of the existing and proposed IVI systems.

	Existing IVI System	Proposed IVI System
**Network Topology**	One-to-one (peer-to-peer)	One-to-many (Star) with IVI-Master
**User Type**	Owner	Owner and users
**Resource Control**	Direct control by user	Control by IVI-Master
**Resource Type**	Personal resource	Personal and shared resources
**Communication Protocols**	MAC/PHY protocols (Bluetooth, ZigBee, etc.)	LWM2M (CoAP, MQTT, HTTP, etc.)

**Table 2 sensors-19-00620-t002:** Parameter values used for simulation.

Parameter	Description	Value
Icomm	Time interval for generation of messages to be sent to a sensor	Random variable with exp(λ=T/3)
Mcon	Size of message for connection	400 bytes
Mcreq	Size of request message for command	420 bytes
Mcres	Size of response message for command	380 bytes
P	Probability that two or more messages will be delivered before transmission	0.15
Tcycle	Period of duty cycle	T/2
Nsensor	Number of IVI Resources	Variable
Ncomm	Number of commands transferred to IVI resources	Random variable with Pois(λ=3 Nsensor)
Ve	Volume of data transferred to IVI resources in the existing scheme	To be calculated
Vp	Volume of data transferred to IVI resources in the proposed scheme	To be calculated

**Table 3 sensors-19-00620-t003:** Specification of the testbed.

	User Device	IVI-Master	IVI Resource
Device Model	*Galaxy Note 5*	*Latte-Panda*	*Raspberry pi 3 B+*
Operating System	Android 7.0 Nougat	Ubuntu 16.04 LTS	Raspbian

**Table 4 sensors-19-00620-t004:** Comparison of bandwidth usages for candidate schemes.

Candidate Scheme	Average (μ)	Standard Deviation (σ)
**Existing Scheme**	1763.6 bytes	564.11 bytes
**Proposed Scheme**	1301 bytes	246.629 bytes

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
