# Peer review of "IoT-Based Resource Control for In-Vehicle Infotainment Services: Design and Experimentation"

_sensors, 2019, doi:10.3390/s19030620_

Round 1
Reviewer 1 Report
The paper is well written and organized. However, for me the work mainly describes a system implementation and it is not suitable for a journal publication. There is no significant contribution, but mainly an implementation of different technologies and protocols. The authors propose a IVI master as an interface between users and resources rather than a directly connection between the ower and the resources. I cannot see a scientific contribution of this proposed scheme compared to the existing scheme. The results are not clear. Did the authors implement the system in a vehicle? What wireless technologies are used in the proposed scheme (CoAP, LWM2M, HTTP are application protocols...)? How the results of figures 19, 20 and 21 were obtained? It is not clear if the authors implemented 50 IVI resources or if it was simulated. There are other references in the literature about infotainment systems in vehicles the authors could cite and use to improve their work.
Author Response
The response file is attached.

Reviewer 2 Report
The paper presents a IoT solution for in-vehicle infotainment services. It is not very clear if the solution is proposed only for non-critical services (like navigation, speakers, etc.) or (as it seems from the description) it also involves critical services (like vehicle sensors, rear view camera, and so on).
If critical services are included, then it seems that a chapter dealing with possible attacks on the system (hacking) and interference with other systems is necessary. Also, more details about the security of the proposed communication should be provided.
Author Response
The response file is attached.

Reviewer 3 Report
This paper proposes a framework of resource control for In-Vehicle Infotainment (IVI) services, based on the Internet-of-Things (IoT) technology for efficiently managing the IVI resources within an automobile. I have some concerns about the paper,
What is the reference of the comparative existing method against the proposed one?
Why do you compare with only one existing method? Why not compare with one more recent existing approach?
Looks like all the results are experimental based. What is the confidence of the results?
Are the results averaged on some trials? Then how many trials?
There is no analytical model to validate the stated results.
Why don't you compare your experimental result with analytical or simulation results?
What is the deviation of the experimental result from the analytical/simulation results?
Author Response
The response file is attached.

Reviewer 4 Report
Comments to the authors:
The authors of this paper present the design of a framework to manage resources in In-Vehicle Infotainment (IVI) systems. The aim is to manage the IVI resources in a vehicle. The proposal is organized in IVI Master to control the IVI resources and the IVI users are classified in owner and users. Also, the resources are identified as personal and shared, which are managed by IVI Master through the use of the Light-Weight Machine-to-Machine (LWM2M) protocol.
This paper is very well written; it is also well structured. The proposed design is adequately described and its operation is validated by using the Wireshark sniffing tool. The testbed uses the Latte-panda board to implement the IVI Master part along with Raspberry PI and Arduino Shield as IVI resources. Such a configuration is depicted in Figure 13 of the paper. Thus, the proposal is compared to a baseline mechanism by measuring the volume of data vs. time, the volume of data vs. the number of IVI resources, as well as the response time vs. the number of IVI resources. Therefore, the proposal outperforms the baseline mechanism in these parameters.
The contribution is clear and sound. Still, it could be improved by much better describing the implementation sketched in Figure 13. There is little detail when pointing out which type of devices where used in the design. The range of models for Latte-panda, Raspberry pi and Arduino Shield is large. Thus, please provide a more insightful description of such a design so a reader can potentially use it as a base for alternative designs or proposals.
Finally, the bandwidth is usually measured in Hz when working on the physical layer or in bits/s when operating on upper layers. However, the unit used to describe bandwidth in the paper does not correspond to these units. Would you consider to replace it with "volume of data" as suggested above? (Or another alternative?)
Author Response
The response file is attached.

Round 2
Reviewer 1 Report
The authors improved the paper. However, for me the contribution is not enough for a journal paper. Maybe the article is suitable for a conference in the area.
Reviewer 3 Report
The authors addressed my concerns in the revised version, hence I recommend to accept the paper.